# Discovery of a new type of topological Weyl fermion semimetal state in Mo$_x$W$_{1-x}$Te$_2$

Ilya Belopolski[1,*], Daniel S. Sanchez[1,*], Yukiaki Ishida[2,*], Xingchen Pan[3,*], Peng Yu[4,*], Su-Yang Xu[1], Guoqing Chang[5,6], Tay-Rong Chang[7], Hao Zheng[1], Nasser Alidoust[1], Guang Bian[1], Madhab Neupane[8], Shin-Ming Huang[5,6], Chi-Cheng Lee[5,6], You Song[9], Haijun Bu[3], Guanghou Wang[3], Shisheng Li[5,6], Goki Eda[5,6,10], Horng-Tay Jeng[7,11], Takeshi Kondo[2], Hsin Lin[5,6], Zheng Liu[4,12,13], Fengqi Song[3], Shik Shin[2] & M. Zahid Hasan[1,14]

The recent discovery of a Weyl semimetal in TaAs offers the first Weyl fermion observed in nature and dramatically broadens the classification of topological phases. However, in TaAs it has proven challenging to study the rich transport phenomena arising from emergent Weyl fermions. The series Mo$_x$W$_{1-x}$Te$_2$ are inversion-breaking, layered, tunable semimetals already under study as a promising platform for new electronics and recently proposed to host Type II, or strongly Lorentz-violating, Weyl fermions. Here we report the discovery of a Weyl semimetal in Mo$_x$W$_{1-x}$Te$_2$ at $x = 25\%$. We use pump-probe angle-resolved photoemission spectroscopy (pump-probe ARPES) to directly observe a topological Fermi arc above the Fermi level, demonstrating a Weyl semimetal. The excellent agreement with calculation suggests that Mo$_x$W$_{1-x}$Te$_2$ is a Type II Weyl semimetal. We also find that certain Weyl points are at the Fermi level, making Mo$_x$W$_{1-x}$Te$_2$ a promising platform for transport and optics experiments on Weyl semimetals.

[1] Laboratory for Topological Quantum Matter and Spectroscopy (B7), Department of Physics, Princeton University, Princeton, New Jersey 08544, USA. [2] The Institute for Solid State Physics (ISSP), University of Tokyo, Kashiwa-no-ha, Kashiwa, Chiba 277-8581, Japan. [3] National Laboratory of Solid State Microstructures, Collaborative Innovation Center of Advanced Microstructures, and Department of Physics, Nanjing University, Nanjing, 210093, China. [4] Centre for Programmable Materials, School of Materials Science and Engineering, Nanyang Technological University, Singapore 639798, Singapore. [5] Centre for Advanced 2D Materials and Graphene Research Centre, National University of Singapore, 6 Science Drive 2, Singapore 117546, Singapore. [6] Department of Physics, National University of Singapore, 2 Science Drive 3, Singapore 117546, Singapore. [7] Department of Physics, National Tsing Hua University, Hsinchu 30013, Taiwan. [8] Department of Physics, University of Central Florida, Orlando, Florida 32816, USA. [9] State Key Laboratory of Coordination Chemistry, School of Chemistry and Chemical Engineering, Collaborative Innovation Center of Advanced Microstructures, Nanjing University, Nanjing 210093, China. [10] Department of Chemistry, National University of Singapore, 3 Science Drive 3, Singapore 117543, Singapore. [11] Institute of Physics, Academia Sinica, Taipei 11529, Taiwan. [12] NOVITAS, Nanoelectronics Centre of Excellence, School of Electrical and Electronic Engineering, Nanyang Technological University, Singapore 639798, Singapore. [13] CINTRA CNRS/NTU/THALES, UMI 3288, Research Techno Plaza, 50 Nanyang Drive, Border X Block, Level 6, Singapore 637553, Singapore. [14] Princeton Institute for Science and Technology of Materials, Princeton University, Princeton, New Jersey 08544, USA. * These authors contributed equally to this work. Correspondence and requests for materials should be addressed to Z.L. (email: z.liu@ntu.edu.sg) or to F.S. (email: songfengqi@nju.edu.cn) or to M.Z.H. (email:mzhasan@princeton.edu).

The recent discovery of the first Weyl semimetal in TaAs has opened a new direction of research in condensed matter physics[1–5]. Weyl semimetals are fascinating because they give rise to Weyl fermions as emergent electronic quasiparticles, have an unusual topological classification closely related to the integer quantum Hall effect, and host topological Fermi arc surface states[6–15]. These properties give rise to many unusual transport phenomena, including negative longitudinal magneto-resistance from the chiral anomaly, an anomalous Hall effect, the chiral magnetic effect, non-local transport and novel quantum oscillations[16–18]. Although many recent works have studied transport properties in TaAs (refs 19–21), transport experiments are challenging because TaAs and its isoelectronic cousins have a three-dimensional crystal structure with irrelevant metallic bands and many Weyl points. As a result, there is a need to discover new Weyl semimetals better suited for transport and optics experiments and eventual device applications.

Recently, the $Mo_xW_{1-x}Te_2$ series has been proposed as a new Weyl semimetal[22–25]. Unlike TaAs, $Mo_xW_{1-x}Te_2$ has a layered crystal structure and is rather widely available as large, high-quality single crystals. Indeed, $MoTe_2$, $WTe_2$ and other transition metal dichalcogenides are already under intense study as a platform for novel electronics[26–30]. Moreover, $Mo_xW_{1-x}Te_2$ offers the possiblity to realize a tunable Weyl semimetal, which may be important for transport measurements and applications. Recently, it was also discovered theoretically that $WTe_2$ hosts a novel type of strongly Lorentz-violating Weyl fermion, or Type II Weyl fermion, long ignored in quantum field theory[23,31–38]. This offers a fascinating opportunity to realize in a crystal an emergent particle forbidden as a fundamental particle in particle physics. There are, moreover, unique transport signatures associated with strongly Lorentz-violating Weyl fermions[23,31–33,39,40]. For all these reasons, there is considerable interest in demonstrating that $Mo_xW_{1-x}Te_2$ is a Weyl semimetal. At the same time, it is important to note that ab initio calculations predict that the Weyl points in $Mo_xW_{1-x}Te_2$ are above the Fermi level[22,24,25]. This makes it challenging to access the Weyl semimetal state with conventional angle-resolved photoemission spectroscopy (ARPES). Recently, we have demonstrated that we can access the unoccupied band structure of $Mo_xW_{1-x}Te_2$ by pump-probe ARPES to the energy range necessary to study the Weyl points and Fermi arcs[41]. As a further consideration, despite the promise of $Mo_xW_{1-x}Te_2$ for transport, if the Weyl points are far from the Fermi level, then the novel phenomena associated with the emergent Weyl fermions and violation of Lorentz invariance will not be relevant to the material's transport properties.

Here we report the discovery of a Weyl semimetal in $Mo_xW_{1-x}Te_2$ at doping $x = 25\%$. We use pump-probe ARPES to study the band structure above the Fermi level and we directly observe two kinks in a surface state band. We interpret the kinks as corresponding to the end points of a topological Fermi arc surface state. We apply the bulk-boundary correspondance and argue that since the surface state band structure includes a topological Fermi arc, $Mo_xW_{1-x}Te_2$ is a Weyl semimetal[42]. The end points of the Fermi arc also allow us to fix the energy and momentum locations of the Weyl points. We find excellent agreement with our ab initio calculation. However, crucially, we find that certain Weyl points have lower binding energy than expected from calculation and, in fact, are located very close to the Fermi level. This unexpected result suggests that our $Mo_{0.25}W_{0.75}Te_2$ samples may be useful to study the unusual transport phenomena of Weyl semimetals and, in particular, those particularly exotic phenomena arising from strongly Lorentz-violating Weyl fermions. Our work also sets the stage for the first tunable Weyl semimetal. Our discovery of a Weyl semimetal in $Mo_xW_{1-x}Te_2$ provides the first Weyl semimetal outside the TaAs family, as well as a Weyl semimetal which may be tunable and easily accessible in transport studies. Taken altogether with calculation, our experimental results further show that we have realized the first Weyl semimetal with Type II, or strongly Lorentz-violating, emergent Weyl fermions.

## Results

**Overview of the crystal and electronic structure**. We first provide a brief background of $Mo_xW_{1-x}Te_2$ and study the band structure below the Fermi level. $WTe_2$ crystallizes in an orthorhombic Bravais lattice, space group $Pmn2_1$ (#31), lattice constants $a = 6.282$ Å, $b = 3.496$ Å and $c = 14.07$ Å, as shown in Fig. 1a (ref. 43). Crucially, the crystal has no inversion symmetry, a requirement for a Weyl semimetal[12]. The crystals we study are flat, shiny, layered and beautiful, see Fig. 1b. The natural cleaving plane is (001), with surface and bulk Brillouin zones, as shown in Fig. 1c. We first consider the overall band structure of $WTe_2$. There are two bands, one electron and one hole pocket, near the Fermi level, both very near the $\Gamma$ point of the bulk Brillouin zone, along the $\Gamma - Y$ line. Although the bands approach each other and Weyl points might be expected to arise where the bands cross, it is now understood that $WTe_2$ is in fact very close to a phase transition between a Weyl semimetal phase and a trivial phase, so that the electronic structure of $WTe_2$ is too fragile to make it a compelling candidate for a Weyl semimetal[22]. Next, we interpolate between ab initio Wannier function-based tight-binding models for $WTe_2$ and $MoTe_2$ to study $Mo_xW_{1-x}Te_2$ at arbitrary $x$ (ref. 22). For a wide range of $x$, we find a robust Weyl semimetal phase[22]. In Fig. 1e,f, we show where the Weyl points sit in the Brillouin zone. They are all located close to $\Gamma$ in the $k_z = 0$ momentum plane. There are two sets of Weyl points, $W_1$ at binding energies $E_B = -0.045$ eV and $W_2$ at $E_B = -0.066$ eV, all above the Fermi level $E_F$. In addition, the Weyl points are almost aligned at the same $k_y = \pm k_W$, although this positioning is not known to be in any way symmetry-protected. We also note that the Weyl cones are all tilted over, corresponding to strongly Lorentz-violating or Type II Weyl fermions, see Fig. 1g (ref. 23). Next, we study a Fermi surface of $Mo_xW_{1-x}Te_2$ at $x = 45\%$ using incident light with photon energy $h\nu = 6.36$ eV, shown in Fig. 1h. We observe two pockets, a palmier-shaped pocket closer to the $\bar{\Gamma}$ point of the surface Brillouin zone and an almond-shaped pocket sitting next to the palmier pocket, further from $\bar{\Gamma}$. The palmier pocket is a hole pocket, while the almond pocket is an electron pocket[41]. We note that we see an excellent agreement between our results and an ab initio calculation of $Mo_xW_{1-x}Te_2$ for $x = 40\%$, shown in Fig. 1i. At the same time, we point out that the electron pocket of the Weyl points is nearly absent in this ARPES spectrum, possibly due to low photoemission cross section at the photon energy used[22,41]. However, as we will see below, we do observe this electron pocket clearly in our pump-probe ARPES measurements, carried out at a slightly different photon energy, $h\nu = 5.92$ eV. On the basis of our calculations and preliminary ARPES results, we expect that the Weyl points sit above the Fermi level, where the palmier and almond pockets approach each other. We also present an $E_B$-$k_x$ spectrum in Fig. 1j, where we see how the plamier and almond pockets nest into each other. We expect the two pockets to chase each other as they disperse above $E_F$, giving rise to Weyl points, see Fig. 1k.

**Unoccupied band structure of $Mo_xW_{1-x}Te_2$**. Next, we show that pump-probe ARPES at probe photon energy $h\nu = 5.92$ eV gives us access to the bulk and surface bands participating in the Weyl semimetal state in $Mo_{0.25}W_{0.75}Te_2$, both below and above $E_F$. In Fig. 2a–c, we present three successive ARPES spectra of $Mo_{0.25}W_{0.75}Te_2$ at fixed $k_y$ near the predicted position of the Weyl

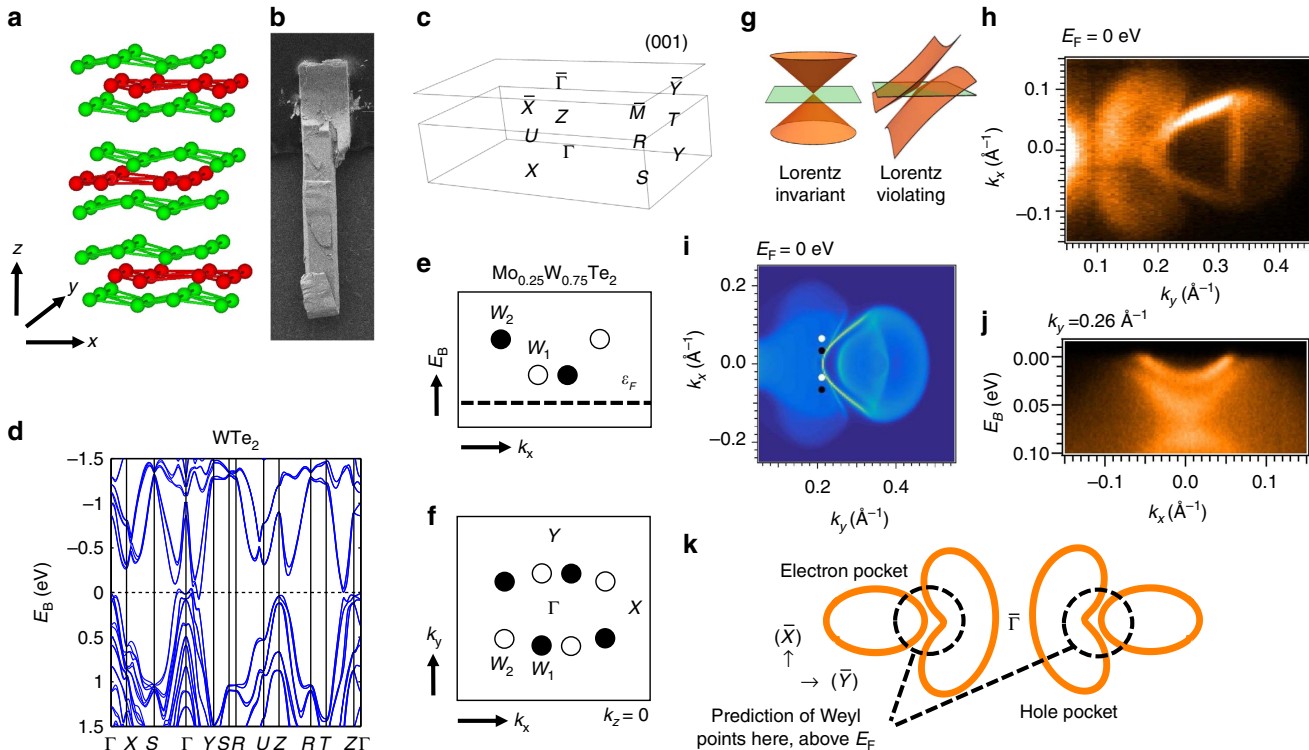

**Figure 1 | Overview of $Mo_xW_{1-x}Te_2$.** (**a**) The crystal structure of the system is layered, with each monolayer consisting of two Te layers (green) and one W/Mo layer (red). (**b**) A wonderful scanning electron microscope image of a typical single crystal of $Mo_xW_{1-x}Te_2$, $x = 45\%$. The layered structure is visible in the small corrugations and breaks in the layers. (**c**) Bulk and (001) surface Brillouin zone, with high-symmetry points marked. (**d**) Bulk band structure of $WTe_2$ along high-symmetry lines. There are two relevant bands near the Fermi level, an electron band and a hole band, both near the $\Gamma$ point and along the $\Gamma - Y$ line, which approach each other near the Fermi level. (**e,f**) On doping by Mo, $Mo_xW_{1-x}Te_2$ enters a robust Weyl semimetal phase[22]. Schematic of the positions of the Weyl points in the bulk Brillouin zone. The opposite chiralities are indicated by black and white circles. Crucially, all Weyl points are above the Fermi level. (**g**) The Weyl cones in $Mo_xW_{1-x}Te_2$ are unusual in that they are all tilted over, associated with strongly Lorentz-violating or Type II Weyl fermions, prohibited in particle physics[23]. (**h**) Fermi surface of $Mo_xW_{1-x}Te_2$ at $x = 45\%$ measured by ARPES at $h\nu = 6.36$ eV, showing a hole-like palmier pocket and an electron-like almond pocket[41]. (**i**) There is an excellent correspondence between our ARPES data and our calculation. Note that the $k_y$ axis on the Fermi surface from ARPES is set by comparison with calculation. (**j**) An $E_B$-$k_x$ cut showing the palmier and almond pockets below the Fermi level. (**k**) In summary, the Fermi surface of $Mo_xW_{1-x}Te_2$ consists of a palmier hole pocket and an almond electron pocket near the $\bar{\Gamma}$ point. The two pockets chase each other as they disperse, eventually intersecting above $E_F$ to give Weyl points.

points. We observe a beautiful, sharp band near $E_F$, whose sharp character suggests that it is a surface band, and broad continua above and below the Fermi level, whose broad character suggests that they are bulk valence and conduction bands. In Fig. 2d–f, we show the same cuts, with guides to the eye to mark the bulk valence and conduction band continua. We also find that we can track the evolution of the bulk valence and conduction bands clearly in our data with $k_y$. Specifically, we see that both the bulk valence and conduction bands disperse toward negative binding energies as we sweep $k_y$ closer to $\bar{\Gamma}$. At the same time, we note that the bulk valence band near $\bar{\Gamma}$ is only visible near $k_x \sim 0$ and drops sharply in photoemission cross-section away from $k_x \sim 0$. In Fig. 2g,h we present a comparison of our ARPES data with an *ab initio* calculation of $Mo_{0.25}W_{0.75}Te_2$ (ref. 22). We also mark the location of the three successive spectra on a Fermi surface in Fig. 2i. We include as well the approximate locations of the Weyl points, as expected from calculation. We find excellent correspondence between both bulk and surface states. We add that we directly observe an additional surface state detaching from the bulk conduction band well above the Fermi level and that we can also match this additional surface state well between our ARPES spectra and calculation. Our pump–probe ARPES results clearly show both the bulk and surface band structure of $Mo_{0.25}W_{0.75}Te_2$, both below and above $E_F$, and with an excellent correspondence with calculation.

**Observation of a topological Fermi arc above the Fermi level.** Now we show that we observe signatures of a Fermi arc in $Mo_{0.25}W_{0.75}Te_2$. We consider the cut shown in Fig. 3a, repeated from Fig. 2b, and we study the surface state. We observe two kinks in each branch, at $E_B \sim -0.005$ eV and $E_B \sim -0.05$ eV. This kink is a smoking-gun signature of a Weyl point[42]. We claim that each kink corresponds to a Weyl point and that the surface state passing through them includes a topological Fermi arc. To show these kinks more clearly, in Fig. 3b, we show a second derivative plot of the spectrum in Fig. 3a. In Fig. 3c we also present a cartoon of the kink in our data, with the positions of the $W_1$ and $W_2$ Weyl points marked. Again, note that although the $W_1$ and $W_2$ are not located strictly at the same $k_y$, we expect the $k_y$ separation to be on the order of $10^{-4}$ Å$^{-1}$ from calculation, so that we can consider them to lie at the same $k_y$ within experimental resolution. We emphasize that from our pump–probe ARPES spectrum, we can directly read off that the energy separation of the Weyl points is $\sim 0.05$ eV and that the $W_1$ are located at $\sim -0.005$ eV. We also present a quantitative analysis of our data, showing a kink. To do this, we fit the surface state momentum distribution curves (MDCs) to a Lorentzian distribution and we plot the train of peaks corresponding to the surface state band. We note that we simultaneously fit the topological surface state, the bulk valence and conduction states, and the trivial surface state above the conduction band. In Fig. 3d we plot

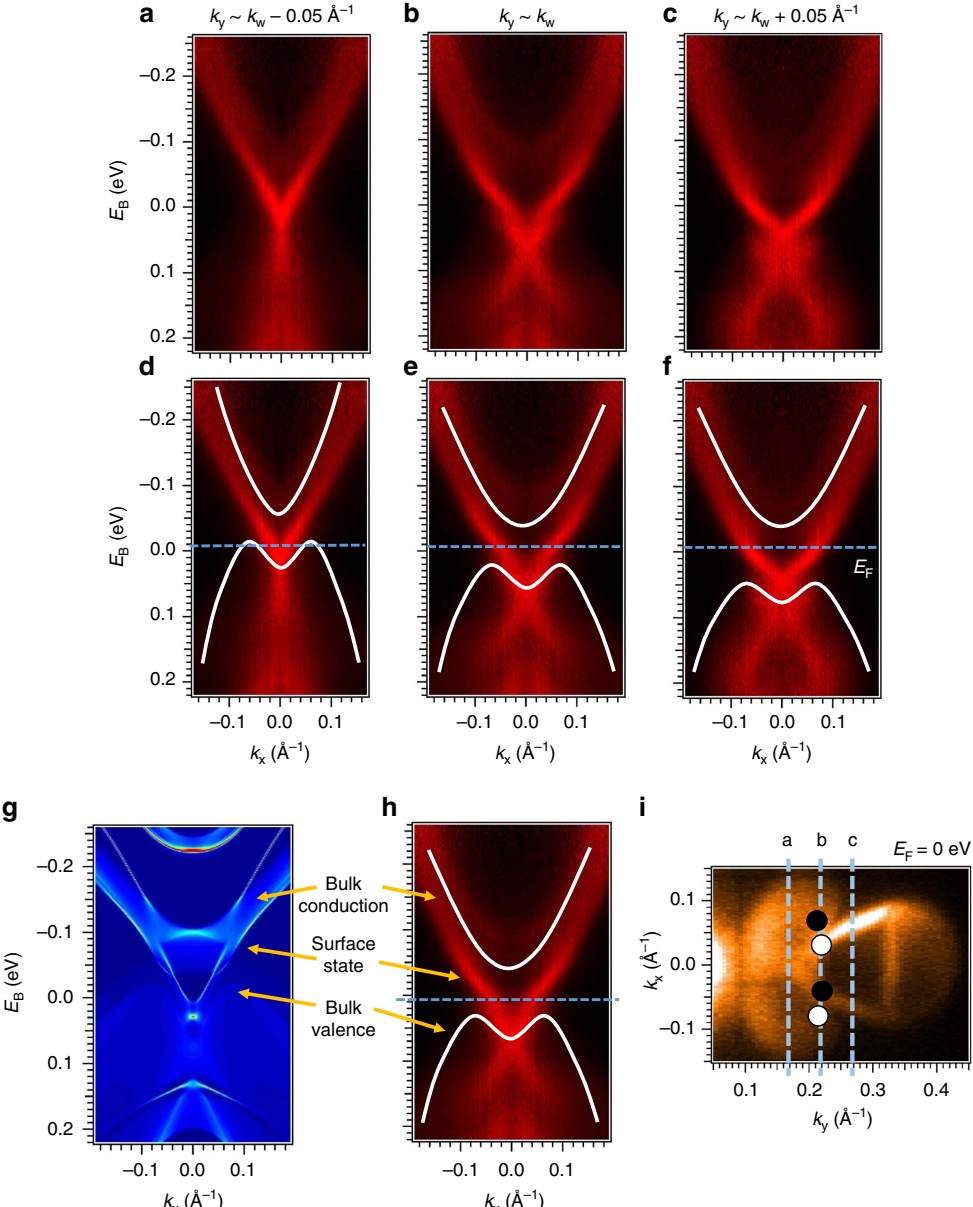

**Figure 2 | Dispersion of the unoccupied bulk and surface states of $Mo_{0.25}W_{0.75}Te_2$.** (**a–c**) Three successive ARPES spectra for $Mo_{0.25}W_{0.75}Te_2$ at fixed $k_y$ near the expected position of the Weyl points, $k_W$, using pump-probe ARPES at probe $hv = 5.92$ eV. A strong pump response allows us to probe the unoccupied states $\sim 0.3$ eV above $E_F$, which is well above the expected $E_{W1}$ and $E_{W2}$. (**d–f**) Same as (**a–c**), but with the bulk valence and conduction band continua marked with guides to the eye. We see that we observe all bulk and surface states participating in the Weyl semimetal state. As expected, both the bulk valence and conduction bands move towards more negative binding energies as $k_y$ moves towards $\bar{\Gamma}$. (**g,h**) Comparison of our calculations with experimental results for $k_y \sim k_W$. As can be seen from panel (**h**), our spectra clearly display all bulk and surface bands of $Mo_{0.25}W_{0.75}Te_2$ relevant for the Weyl semimetal state, both below and above $E_F$, and with excellent agreement with the corresponding calculation in panel **g**. (**i**) The locations of the cuts in (**a–c**).

the resulting band dispersions in white and observe an excellent fit to our spectrum. Next, we define a kink as a failure of the train of Lorentzian maxima to fit to a quadratic band. In particular, over a small energy and momentum window, any band should be well-characterized by a quadratic fit, so the failure of such a fit in a narrow energy window implies a kink. After fitting the topological surface state to a quadratic polynomial we find two mismatched regions, marked in Fig. 3e, demonstrating two kinks. For comparison, we plot the energy positions of the $W_1$ and $W_2$ as read off directly from Fig. 3a. We find an excellent agreement between the qualitative and quantitative analysis, although we note that the fit claims that the $W_2$ kink is lower in energy. To

illustrate the success of the Lorentzian fit, in Fig. 3f,g we present two representative MDCs at energies indicated by the green arrows. We see that the Lorentzian distributions provide a good fit and take into account all bands observed in our spectra. The raw data, the second derivative plots and the Lorentzian fitting all show two kinks, providing a strong signature of Fermi arcs.

To show that we have observed a topological Fermi arc, we compare our experimental observation of two surface state kinks with a numerical calculation of $Mo_{0.25}W_{0.75}Te_2$. In Fig. 4a,b, we mark the energies of the Weyl points as well as the band minimum of the surface state in our ARPES spectrum and in calculation. We see that the energy difference between the Weyl

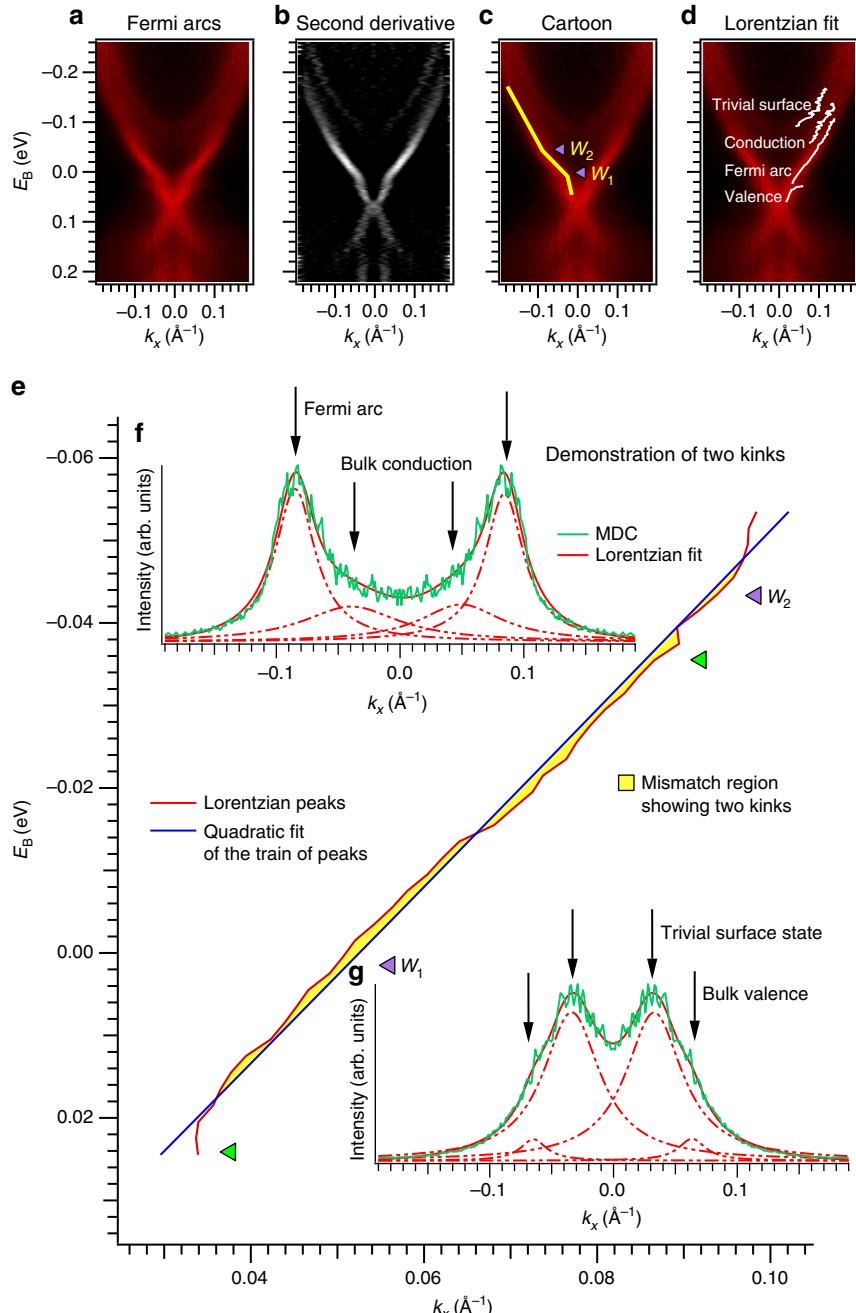

**Figure 3 | Direct experimental observation of Fermi arcs in Mo$_{0.25}$W$_{0.75}$Te$_2$.** (**a**) To establish Fermi arcs in Mo$_{0.25}$W$_{0.75}$Te$_2$ we focus on the spectrum shown in Fig. 2b, with $k_y \sim k_W$. We observe two kinks in the surface state, at $E_B \sim -0.005$ eV and $E_B \sim -0.05$ eV. (**b**) The kinks are easier to see in a second-derivative plot of panel **a**. (**c**) Same as panel **a**, but with a guide to the eye showing the kinks. The Weyl point projections are at the locations of the kinks. The surface state with the kinks is a topological Fermi arc. (**d**) To further confirm a kink, we fit Lorentzian distributions to our data. We capture all four bands in the vicinity of the kinks: the bulk conduction and valence states, the topological surface state and an additional trivial surface state merging into the conduction band at more negative $E_B$. We define a kink as a failure of a quadratic fit to a band. We argue that for a small energy and momentum window, any band should be well-characterized by a quadratic fit and that the failure of such a fit shows a kink. (**e**) By matching the train of Lorentzian peaks of the topological surface state (red) to a quadratic fit (blue) we find two mismatched regions (shaded in yellow), showing two kinks. The purple arrows show the location of the Weyl points, taken from panel **c**, and are consistent with the kinks we observe by fitting. (**f,g**) Two characteristic MDCs at energies indicated by the green arrows in panel **e**. We see that the Lorentzian distributions provide a good fit and capture all bands observed in our spectra.

points is $\sim 0.02$ eV in calculation but $\sim 0.05$ eV in experiment. Moreover, the band minimum $E_{min}$ is at $\sim E_F$ in calculation, but at $E_B \sim 0.06$ eV in experiment. The difference in $E_{min}$ suggests either that our sample is electron-doped or that the $k_y$ position of the Weyl points differs in experiment and theory. Next, crucially, we observe that, in disagreement with calculation, the $W_1$ are

located only $\sim 0.005$ eV above $E_F$. This suggests that the Weyl points and Fermi arcs in our Mo$_{0.25}$W$_{0.75}$Te$_2$ samples may be accessible in transport. This result is particularly relevent because MoTe$_2$, WTe$_2$ and other transition metal dichalcogenides are already under study as platforms for novel electronics[26–30]. Since the Weyl points of Mo$_x$W$_{1-x}$Te$_2$ may be at the Fermi level, it is

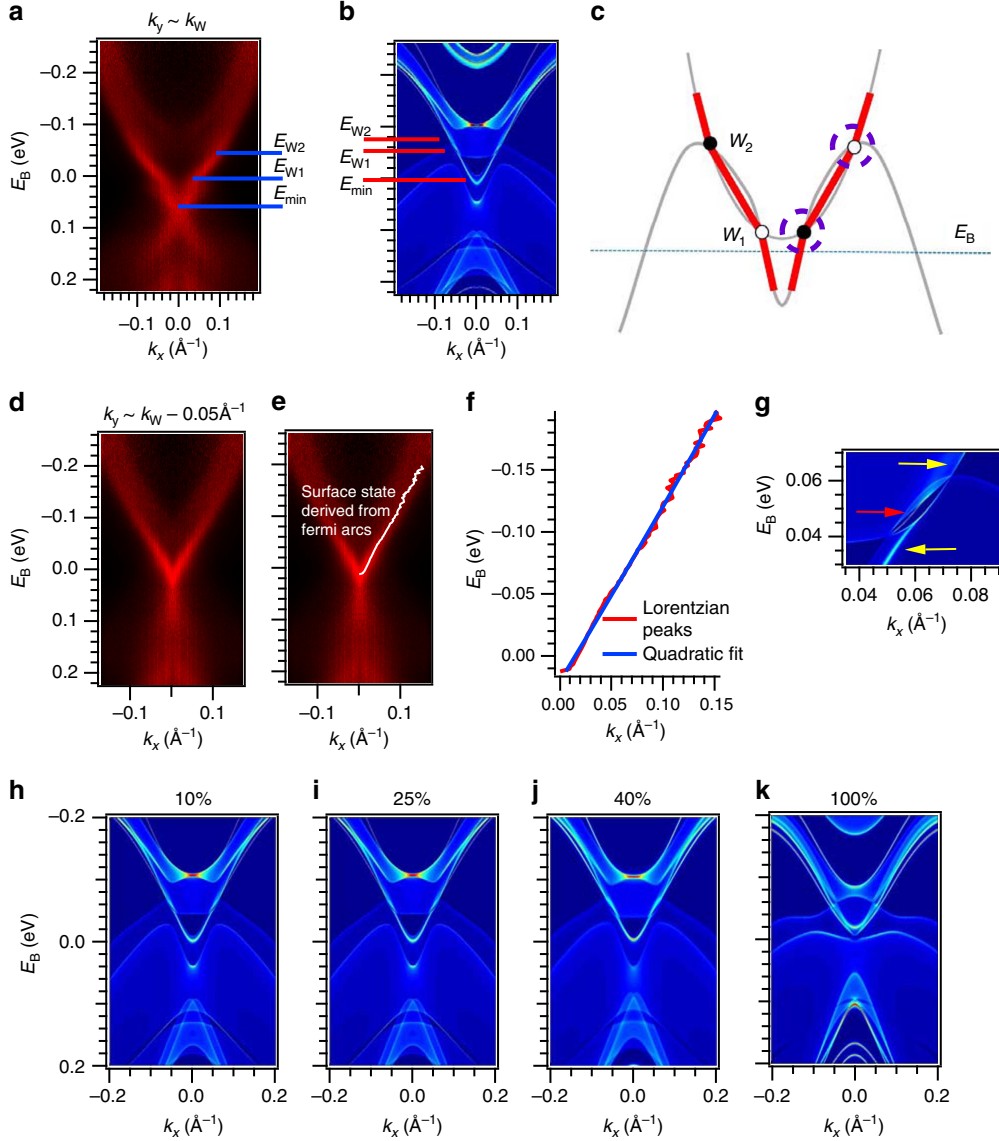

**Figure 4 | Demonstration of a Weyl semimetal in Mo$_x$W$_{1−x}$Te$_2$.** (**a**) The same spectrum as Fig. 3a but with the energies $E_{W1}$, $E_{W2}$, $E_{min}$ marked. (**b**) The same energies marked in an *ab initio* calculation of Mo$_{0.25}$W$_{0.75}$Te$_2$. We note that this cut is not taken at fixed $k_y$–$k_W$. Instead, we cut along the exact line defined by $W_1$ and $W_2$ in the surface Brillouin zone. Since $k_y^{W1}$ is exceedingly close to $k_y^{W2}$, this cut essentially corresponds to our experimental data. The Weyl points are ∼0.05 eV separated in energy in our data, compared with ∼0.02 eV in calculation. In addition, crucially, the $W_1$ are lower in energy than we expect from calculation and in fact are located only ∼0.005 eV above $E_F$. (**c**) A cartoon of our interpretation of our experimental results. We observe the surface state (red) with a kink at the locations of the Weyl points (black and white circles). Each surface state consists of a Fermi arc (middle red segment) and two trivial surface states which merge with bulk bands near the location of the Weyl points. We observe certain portions of the bulk bands (grey), but not the bulk Weyl cones. (**d**) The same spectrum as Fig. 2c, at $k_y$ shifted toward $\bar{\Gamma}$. (**e,f**) A Lorentzian fit of the surface state and a quadratic fit to the train of peaks, showing no evidence of a kink. This is precisely what we expect from a cut away from the Weyl points. (**g**) A close-up of the band inversion, showing a Fermi arc (red arrow) which connects the Weyl points and trivial surface states (yellow arrows) from above and below which merge with the bulk bands in the vicinity of the Weyl points. (**h–k**) Composition dependence of Mo$_x$W$_{1−x}$Te$_2$ from first principles, showing that the separation of the Weyl points increases with $x$. Our observation of a Weyl semimetal in Mo$_{0.25}$W$_{0.75}$Te$_2$ sets the stage for the first tunable Weyl semimetal in Mo$_x$W$_{1−x}$Te$_2$.

possible that transport measurements may detect a signature of the strongly Lorentz-violating Weyl fermions or other unusual transport phenomena associated with Weyl semimetals in Mo$_x$W$_{1−x}$Te$_2$. We summarize our results in Fig. 4c. We directly observe, above the Fermi level, a surface state with two kinks (shown in red). By comparing our results with *ab initio* calculation, we confirm that the kinks correspond to Weyl points. Furthermore, the excellent agreement of our experimental results with calculation shows that we have realized the first Type II Weyl semimetal.

**Limits on directly observing Type II Weyl cones.** So far we have studied the surface states of Mo$_x$W$_{1−x}$Te$_2$ and we have argued that Mo$_x$W$_{1−x}$Te$_2$ is a Weyl semimetal because we observe a topological Fermi arc surface state. However, topological Fermi arcs cannot strictly distinguish between bulk Weyl cones that are of Type I or Type II. While the excellent agreement with calculation suggests that Mo$_x$W$_{1−x}$Te$_2$ is a Type II Weyl semimetal, we might ask if we can directly observe a Type II Weyl cone in Mo$_x$W$_{1−x}$Te$_2$ by ARPES. This corresponds to observing the two branches of the bulk Weyl cone, as indicated by the purple dotted

circles in Fig. 4c. We reiterate that one crucial obstacle in observing a Type II Weyl cone is that all the recent calculations on $WTe_2$, $Mo_xW_{1-x}Te_2$ and $MoTe_2$ predict that all Weyl points are above the Fermi level[22–25]. As we have seen, using pump-probe ARPES, we are able to measure the unoccupied band structure and show a Fermi arc. However, in our pump-probe ARPES measurements, we find that the photoemission cross-section of the bulk bands is too weak near the Weyl points. At the same time, our calculations suggest that for a reasonable quasiparticle lifetime and spectral linewidth, it may be difficult to resolve the two branches of the Weyl cone. We conclude that it is challenging to directly access the Type II Weyl cones in $Mo_xW_{1-x}Te_2$.

**Considerations regarding trivial surface states.** One obvious concern in the interpretation of our experimental result is that we observe two kinks in the surface state, but we expect a disjoint segment based on topological theory. In particular, all calculations show that all Weyl points in $Mo_xW_{1-x}Te_2$ have chiral charge $\pm 1$ (refs 22–25). However, our observation of a kink suggests that there are two Fermi arcs connecting to the same Weyl point, which requires a chiral charge of $\pm 2$. To resolve this contradiction, we study the calculation of the surface state near the Weyl points, shown in Fig. 4g. We observe, as expected, a Fermi arc (red arrow) connecting the Weyl points. However, at the same time, we see that trivial surface states (yellow arrows) from above and below the band crossing merge with the bulk bands in the vicinity of the Weyl points. As a result, there is no disjoint arc but rather a large, broadband surface state with a ripple arising from the Weyl points. We can imagine that this broadband surface state exists even in the trivial phase. Then, when the bulk bands cross and give rise to Weyl points, a Fermi arc is pulled out from this broadband surface state. At the same time, the remainder of the broadband surface state survives as a trivial surface state. In this way, the Fermi arc is not disjoint but shows up as a ripple. We observe precisely this ripple in our ARPES spectra of $Mo_{0.25}W_{0.75}Te_2$.

As a further check of our analysis, we perform a Lorentzian fit of an ARPES spectrum at $k_y$ shifted away from the Weyl points, shown in Fig. 4d, the same cut as Fig. 2c. We show the Lorentzian fit in Fig. 4e and a quadratic fit to the train of peaks in Fig. 2f. In sharp contrast to the result for $k_y \approx k_W$, there is no ripple in the spectrum and the quadratic provides an excellent fit. This result is again consistent with our expectation that we should observe a ripple only at $k_y$ near the Weyl points. Our results also set the stage for the realization of the first tunable Weyl semimetal in $Mo_xW_{1-x}Te_2$. As we vary the composition, we expect to tune the relative separation of the Weyl points and their position in energy relative to $E_F$. In Fig. 4h–k, we present a series of calculations of $Mo_xW_{1-x}Te_2$ for $x = 10$, 25, 40 and 100%. We see that the separation of the Weyl points increases with $x$ and that the $W_1$ approach $E_F$ for larger $x$. We propose that a systematic composition dependence can demonstrate the first tunable Weyl semimetal in $Mo_xW_{1-x}Te_2$.

## Discussion

We have demonstrated a Weyl semimetal in $Mo_xW_{1-x}Te_2$ by directly observing kinks and a Fermi arc in the surface state band structure. Taken together with calculation, our experimental data show that we have realized the first Type II Weyl semimetal, with strongly Lorentz-violating Weyl fermions. We point out that in contrast to concurrent works on the Weyl semimetal state in $MoTe_2$ (refs 44–52), we directly access the unoccupied band structure of $Mo_xW_{1-x}Te_2$ and directly observe a Weyl semimetal with minimal reliance on calculation. In particular, our observation of a surface state kink at a generic point in the

surface Brillouin zone requires that the system be a Weyl semimetal[42]. The excellent agreement with calculation serves as an additional, independent check of our experimental results. We also reiterate that unlike $MoTe_2$, $Mo_xW_{1-x}Te_2$ opens the way to the realization of the first tunable Weyl semimetal. Lastly, we note that $MoTe_2$ is complicated because it is near a critical point for a topological phase transition. Indeed, one recent theoretical work[24] shows that $MoTe_2$ has four Weyl points, while another[25] finds eight Weyl points. This is, moreover, similar to the case of $WTe_2$, which is near the critical point for a transition between eight Weyl points and zero Weyl points. By contrast, $Mo_xW_{1-x}Te_2$ sits well within the eight Weyl point phase for most $x$, as confirmed explicitly here and by calculation in ref. 22. The stability of the topological phase of $Mo_xW_{1-x}Te_2$ simplifies the interpretation of our data. By directly demonstrating a Weyl semimetal in $Mo_xW_{1-x}Te_2$, we provide not only the first Weyl semimetal beyond the TaAs family, but the first Type II Weyl semimetal, as well as a Weyl semimetal which may be tunable and which may be more accessible for transport and optics studies of the fascinating phenomena arising from emergent Weyl fermions in a crystal.

## Methods

**Pump-probe ARPES.** Pump-probe ARPES measurements were carried out using a hemispherical Scienta R4000 analyser and a mode-locked Ti:Sapphire laser system that delivered 1.48 eV pump and 5.92 eV probe pulses at a repetition rate of 250 kHz (ref. 53). The time and energy resolution were 300 fs and 15 meV, respectively. The spot diameters of the pump and probe lasers at the sample were 250 and 85 μm, respectively. Measurements were carried out at pressures $< 5 \times 10^{-11}$ Torr and temperatures $\sim 8$ K.

**Sample growth.** Single crystals of $Mo_xW_{1-x}Te_2$ were grown using a chemical vapor transport technique with iodine as the transport agent. Stoichiometric Mo, W and Te powders were ground together and loaded into a quartz tube with a small amount of I. The tube was sealed under vacuum and placed in a two-zone furnace. The hot zone was maintained at 1,050 °C for 2 weeks and the cold zone was maintained at 950 °C. The dopant distribution is not uniform particularly near the crystal surface. The composition of the selected sample was determined by an energy dispersive spectroscopy measurement with a scanning electron microscope.

***Ab initio* calculations.** The *ab initio* calculations were based on the generalized gradient approximation[54] using the full-potential projected augmented wave method[55,56] as implemented in the VASP package[57]. Experimental lattice constants were used for both $WTe_2$ (ref. 58) and $MoTe_2$. A 15 × 11 × 7 Monkhorst-Pack $k$-point mesh was used in the computations. The spin-orbit coupling effects were included in calculations. To calculate the bulk and surface electronic structures, we constructed first-principles tight-binding model Hamilton by projecting onto the Wannier orbitals[59–61], which use the VASP2WANNIER90 interface[62]. We used W $d$ orbitals, Mo $d$ orbitals, and Te $p$ orbitals to construct Wannier functions and without perform the procedure for maximizing localization. The electronic structure of the $Mo_xW_{1-x}Te_2$ samples with finite doping was calculated by a linear interpolation of tight-binding model matrix elements of $WTe_2$ and $MoTe_2$. The surface states were calculated from the surface Green's function of the semi-infinite system[63,64].

**Data availability.** The data relevant to the findings of this study are available from the corresponding authors on reasonable request.

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

## Acknowledgements

I.B. thanks Yun Wu, Daixiang Mou, Lunan Huang and Adam Kaminski for collaboration on the conventional laser ARPES measurements and insightful discussions on $Mo_xW_{1-x}Te_2$ and $MoTe_2$. Work at Princeton University by I.B., D.S.S., S.-Y.X., H.Z., N.A. and G.B. is supported by the US Department of Energy under Basic Energy Sciences Grant No. DOE/BES DE-FG-02-05ER46200 and Princeton University funds (M.Z.H.). Additional support is provided by the Gordon and Betty Moore Foundation under Grant No. GBMF4547. I.B. acknowledges the support of the US National Science Foundation GRFP. Y.I. is supported by the Japan Society for the Promotion of Science, KAKENHI 26800165. T.-R.C. and H.-T.J. were supported by the National Science Council, Taiwan. H.-T.J. also thanks the National Center for High-Performance Computing, Computer and Information Network Center National Taiwan University, and National Center for Theoretical Sciences, Taiwan, for technical support. M.N. is supported by start-up funds from the University of Central Florida. X.C.P., Y.S., H.J.B., G.H.W and F.Q.S. thank the National Key Projects for Basic Research of China (grant nos. 2013CB922100, 2011CB922103), the National Natural Science Foundation of China (grant nos. 91421109, 11522432 and 21571097) and the NSF of Jiangsu province (No. BK20130054). This work is also financially supported by the Singapore National Research Foundation under NRF award no. NRF-NRFF2013-03 (H.L.); NRF Research Fellowship (NRF-NRFF2011-02) and the NRF Medium-sized Centre program (G.E.) and NRF RF award no. NRF-RF2013-08, the start-up funding from Nanyang Technological University (M4081137.070) (Z.L.).

## Author contributions

Pump-probe ARPES measurements were carried out by I.B. and D.S.S. with assistance from S.-Y.X., N.A., G.B. and M.N. with guidance from M.Z.H. The pump-probe ARPES system was maintained, calibrated and optimized by Y.I. with assistance from T.K. and under the direction of S.S. Samples were grown by X.P., P.Y., Z.L. and F.S. with assistance from Y.S., H.B. and G.W. G.C., T.-R.C., S.-M.H., C.-C.L., H.-T.J. and H.L. carried out numerical calculations with experimental data on lattice constants provided by S.L. and G.E. The manuscript was written primarily by I.B., D.S.S., Y.I. and S.-Y.X., with valuable insights and comments from all authors. M.Z.H. provided overall direction, planning and guidance for the project.

## Additional information

**Competing financial interests:** The authors declare no competing financial interests.

