## [Peer Review File · Nature Communications]

REVIEWERS' COMMENTS:

Reviewer #1 (Remarks to the Author):

I find that the authors have answered my previous comments in a satisfactory manner. I believe the paper may be published in Nature Communications in its present form.

Reviewer #3 (Remarks to the Author):

I find that the authors have responded nicely to my concerns. Instead of observing the Weyl nodes that is unfortunately limited by both of the apparatus and the material, they have assigned the number and the location of Weyl points from the observed kinks in the observed energy dispersion. This has successfully been achieved due to the higher angular resolutions utilizing the low energy photons. I think that thanks to nature the photoemission signal was not sacrificed by the fatal photoionization cross section for surface and bulk states. Then I can now recommend this work to be published in Nature Communications after the following minor revisions will be made.

p.5: "However, as we discuss below, we can observe We expect the Weyl points to sit above the Fermi level, where the palmier and almond pockets approach each other."
=> Please correct this sentence in a proper way.

Fig.2j is still confusing. It seems that a, b and c in Fig.2j must be $c \Rightarrow b \Rightarrow a$ from the left. Otherwise the dispersion features in Fig.2a-c contradict with those in Fig.2j. I mean that the size of fermi wave vector along k_x shrinks from a to c in Fig.2, while it widens in Fig.2j.

Fig.4g: "A close-up of the band inversion, showing a Fermi arc which connects the Weyl points and trivial surface states from above and below which merge with the bulk bands in the vicinity of the Weyl points."
=> I do not clearly see where is trivial surface states in the panel g.

Re: NCOMMS-16-17861-T, Discovery of a new type of Weyl semimetal state in $\text{Mo}_x\text{W}_{1-x}\text{Te}_2$, by I. Belopolski, *et al.*

We thank the editors for their consideration and we are excited that they have concluded that our work likely merits publication in *Nature Communications*.

Below, we address the remaining critiques of the reviewers point by point.

REPORT OF REVIEWER 1

Reviewer 1: I find that the authors have answered my previous comments in a satisfactory manner. I believe the paper may be published in *Nature Communications* in its present form.

Authors: We thank the reviewer for her/his useful comments and we are excited that the reviewer feels our work is ready for *Nature Communications*.

REPORT OF REVIEWER 3

Reviewer 3: I find that the authors have responded nicely to my concerns. Instead of observing the Weyl nodes that is unfortunately limited by both of the apparatus and the material, they have assigned the number and the location of Weyl points from the observed kinks in the observed energy dispersion. This has successfully been achieved due to the higher angular resolutions utilizing the low-energy photons. I think that thanks to nature the photoemission signal was not sacrificed by the fatal photoionization cross section for surface and bulk states.

Authors: We are happy that the reviewer is satisfied by our response and we again thank the reviewer for her/his valuable remarks. The reviewer accurately summarizes the key ideas of our manuscript as far as the Weyl points, Fermi arcs and low-energy photoemission technique are concerned.

Reviewer 3: Then, I can now recommend this work to be published in *Nature Communications* after the following minor revisions will be made.

Authors: We are excited that the reviewer feels our work is almost ready for publication in *Nature Communications*.

Reviewer 3: p.5: “However, as we discuss below, we can observe We expect the Weyl points to sit above the Fermi level, where the palmier and almond pockets approach each other.”
 \Rightarrow Please correct this sentence in a proper way.

Authors: We thank the reviewer for carefully reading our text. We apologize for the error in typing. We have corrected the sentence to: “We expect the Weyl points to sit above the Fermi level, where the palmier and almond pockets approach each other.” We note that there is no change to the meaning of the text.

Reviewer 3: Fig. 2j is still confusing. It seems that a, b and c in Fig. 2j must be $c \Rightarrow b \Rightarrow a$ from the left. Otherwise the dispersion features in Fig. 2a-c contradict with those in Fig. 2j. I mean that the size of Fermi wave vector along k_x shrinks from a to c in Fig. 2, while it widens in Fig. 2j.

Authors: We thank the reviewer for the good catch. We apologize for the error in the figure panel placement. We have swapped panels (a) with (c) and (d) with (f) in Fig. 2. We emphasize that there is no change in our intended meaning. In fact, the labels above panels (a) and (c) indicated the correct value for k_y and remain unchanged. We just made a mistake and had the panels swapped.

Reviewer 3: Fig. 4g: “A close-up of the band inversion, showing a Fermi arc which connects the Weyl points and trivial surface states from above and below which merge with the bulk bands in the vicinity of the Weyl points.” \Rightarrow I do not clearly see where is trivial surface states in the panel g.

Authors: We thank the reviewer for the remark. We have added some red and yellow arrows in Fig. 4g to indicate the Fermi arcs and trivial surface states. Again, we note that we are not changing in any way our results or interpretation, just improving the presentation and clarity.